# Polydopamine-Coated Co_3_O_4_ Nanoparticles as an Efficient Catalase Mimic for Fluorescent Detection of Sulfide Ion

**DOI:** 10.3390/bios12111047

**Published:** 2022-11-19

**Authors:** Trung Hieu Vu, Phuong Thy Nguyen, Moon Il Kim

**Affiliations:** Department of BioNano Technology, Gachon University, 1342 Seongnamdae-ro, Sujeong-gu, Seongnam 13120, Republic of Korea

**Keywords:** polydopamine coating, cobalt oxide nanoparticles, catalase-like nanozyme, sulfide ion detection, fluorescent biosensors

## Abstract

Surface engineering of nanozymes has been recognized as a potent strategy to improve their catalytic activity and specificity. We synthesized polydopamine-coated Co_3_O_4_ nanoparticles (PDA@Co_3_O_4_ NPs) through simple dopamine-induced self-assembly and demonstrated that these NPs exhibit catalase-like activity by decomposing H_2_O_2_ into oxygen and water. The activity of PDA@Co_3_O_4_ NPs was approximately fourfold higher than that of Co_3_O_4_ NPs without PDA, possibly due to the additional radical scavenging activity of the PDA shell. In addition, PDA@Co_3_O_4_ NPs were more stable than natural catalase under a wide range of pH, temperature, and storage time conditions. Upon the addition of a sample containing sulfide ion, the activity of PDA@Co_3_O_4_ NPs was significantly inhibited, possibly because of increased mass transfer limitations via the absorption of the sulfide ion on the PDA@Co_3_O_4_ NP surface, along with NP aggregation which reduced their surface area. The reduced catalase-like activity was used to determine the levels of sulfide ion by measuring the increased fluorescence of the oxidized terephthalic acid, generated from the added H_2_O_2_. Using this strategy, the target sulfide ion was sensitively determined to a lower limit of 4.3 µM and dynamic linear range of up to 200 µM. The fluorescence-based sulfide ion assay based on PDA@Co_3_O_4_ NPs was highly precise when applied to real tap water samples, validating its potential for conveniently monitoring toxic elements in the environment.

## 1. Introduction

Sulfide ions (S^2−^), which are among the most harmful contaminants, are extensively released into aqueous environments through various agricultural and industrial processes. These ions can exist in the human body and take part in the antioxidant process in liver and lung, or act as vasodilators [1,2,3]. Importantly, an imbalance in S^2−^ levels has been implicated in various diseases such as Alzheimer’s disease, Down’s syndrome, hyperglycemia, and liver cirrhosis [4,5,6,7]. Abnormally high levels of S^2−^ can directly threaten both the ecological environment and human health. To date, many methods for detecting S^2−^ have been developed, such as gas chromatography [8,9], titration [10], extraction [11], colorimetric [12,13], electrochemical [14], and fluorometric assays [15]. However, these methods are often time-consuming because of the sample pre/post treatments required, involved complicated assay procedures, and need for qualified operators [16]. Thus, more rapid, convenient, selective, sensitive, and reliable analytical methods for S^2−^ detection are urgently needed.

Catalase, which is commonly found in most aerobic organisms, plays a crucial role in protecting cells against oxidative damage, by decomposing H_2_O_2_ into non-harmful O_2_ and H_2_O [17]. Similar to other natural enzymes, catalase is unstable under harsh conditions, is costly to produce and purify, and is difficult to recycle. To overcome these limitations, studies aimed at developing an appropriate catalase mimic from nanomaterials exhibiting enzyme-like activities (nanozymes) have gained attention. To date, several types of catalase-mimicking nanozymes have been reported, such as cerium oxide nanoparticles [18], iron oxide NPs [19], and cobalt oxide (Co_3_O_4_) NPs [20,21]. Among these, Co_3_O_4_ NPs showed morphology-dependent catalase-like activity; however, few studies have reported their surface engineering, which is an efficient approach for engineering nanozymes with improved activity [22,23].

Herein, we developed polydopamine (PDA)-coated Co_3_O_4_ NPs (PDA@Co_3_O_4_ NPs) as highly active catalase-mimicking nanozymes and applied these NPs in fluorescent detection of S^2−^. PDA can be produced through self-polymerization of dopamine (DA) without the involvement of any organic solvent, yielding unique adhesion properties based on its active catechol and alkane groups, to facilitate substrate adsorption and product desorption in diverse reactions [24,25,26]. The synthesized PDA@Co_3_O_4_ NPs showed high catalase-like activity in degrading H_2_O_2_, which inhibited the formation of extremely fluorescent 2-hydroxy terephthalic acid, to decrease the fluorescent signal. Interestingly, S^2−^ in the sample may interact with PDA@Co_3_O_4_ NPs, resulting in decreased catalase-like activity via increased substrate transfer limitations and a decreased available surface area for catalytic events. We detected the target S^2−^ in a PDA@Co_3_O_4_ NPs-based fluorescent assay and investigated various analytical characteristics, such as selectivity, sensitivity, stability, and practical utility along with the detection precision.

## 2. Materials and Methods

### 2.1. Reagents and Materials

Cobalt (II) sulfate heptahydrate (CoSO_4_·7H_2_O), DA hydrochloride, trizma hydrochloride (Tris-HCl), terephthalic acid (TA), sodium acetate (NaAc), sodium borohydride (NaBH_4_), and sodium sulfide (Na_2_S) were purchased from Sigma-Aldrich (St. Louis, MO, U.S.A.). Hydrogen peroxide was obtained from Samchun Chemical (Seoul, Korea). All solutions were prepared in deionized water purified using a Milli-Q Purification System (Millipore, Billerica, MA, U.S.A.).

### 2.2. Synthesis and Characterization of PDA@Co_3_O_4_ NPs

PDA@Co_3_O_4_ NPs were synthesized following a previously reported method for DA-induced self-assembly with some modifications [25]. First, 250 mg CoSO_4_∙7H_2_O was dissolved in 50 mL of Tris-HCl buffer (pH 8.5) at 25 °C. DA (1 mg/mL) was added to the solution, followed by sonication at a frequency of 40 kHz and power of 160 W for 20 min. Next, 2 mL of 50 mM NaBH_4_ was added, and the mixture was incubated at 60 °C for 4 h. The pellet was collected by a centrifugation (8000× *g*, 8 min), followed by freeze-drying to obtain the resulting black powders. Bare Co_3_O_4_ NPs without a PDA shell were synthesized using the same procedures except that H_2_O was added instead of DA. 

The synthesized materials were analyzed using scanning electron microscopy (SEM) and transmission electron microscopy (TEM), using a field emission scanning electron microscope (Hitachi S-4700, Tokyo, Japan) and transmission electron microscope (FEI Tecnai, OR), respectively. Elemental composition was analyzed using an energy-dispersive spectrometer (EDX) (Bruker, Billerica, MA, U.S.A.). For SEM analyses, the suspension of sonicated NPs was dried on a silicon wafer. For TEM analyses, 5 μL of the suspension of sonicated NPs was dropped onto a carbon-coated copper TEM grid (Electron Microscopy Sciences, Hatfield, U.K.) followed by drying overnight at room temperature (RT). Fourier transform infrared (FT-IR) spectra of the NPs were obtained using an FT-IR spectrophotometer (FT/IR-4600, JASCO, Tokyo, Japan). X-ray diffraction (XRD) and X-ray photoelectron spectroscopy (XPS) were conducted using an X-ray diffractometer (D/MAX-2500, Rigaku Corporation, Tokyo, Japan) and XPS reader (Gemini, Molecular Devices, Sunnyvale, CA, U.S.A.), respectively. The size distribution of the NPs was determined using dynamic light scattering (DLS) (Zetasizer, Malvern Instruments, Malvern, U.K.).

### 2.3. Evaluation of Enzyme-Like Activity of PDA@Co_3_O_4_ NPs 

Catalase-like activity of PDA@Co_3_O_4_ NPs was examined by using TA as a fluorescent probe in the presence of H_2_O_2_. In a standard assay, PDA@Co_3_O_4_ NPs or bare Co_3_O_4_ NPs (both at 100 μg/mL), H_2_O_2_ (20 mM), and TA (0.625 mM) were incubated in NaAc buffer solution (0.1 M, pH 6.0) for 20 min at RT under UV irradiation at 254 nm using a UV transilluminator (Core-Bio System, Seoul, Korea). The fluorescent signal was monitored using a microplate reader (Synergy H1, BioTek, Winooski, VT, U.S.A.) at excitation and emission wavelengths of 315 and 420 nm, respectively. The effects of the PDA@Co_3_O_4_ NP concentration on their catalytic activity were examined following the same procedures but with varying concentrations of PDA@Co_3_O_4_ NPs (0, 6.25, 12.5, 25, 50, 100, 150, 200, 250, and 300 μg/mL). The effects of the reaction pH and temperature on the catalytic activity of PDA@Co_3_O_4_ NPs and natural free catalase were evaluated over wide pH (3.0–9.0) and temperature (4–80 °C) ranges. Stabilities of PDA@Co_3_O_4_ NPs and free catalase were investigated by incubating them for 8 h in NaAc buffer at various pH values (3.0–10.0) at RT or temperatures (4–80 °C) at pH 6.0. The long-term stabilities of PDA@Co_3_O_4_ NPs and free catalase were also evaluated by incubating them in NaAc buffer (pH 6.0) under static conditions at RT. The initial activities were determined by measuring the fluorescence intensity under standard assay conditions, and relative activity (%) was determined by calculating the ratio of residual to initial activity. Fluorescent images were acquired using a fluorescence imaging system (Kodak, Tokyo, Japan). 

Steady-state kinetic parameters of the catalase-like activity of PDA@Co_3_O_4_ NPs were determined based on oxygen production measured using a dissolved oxygen meter (Eutech DO 6+, Thermo Fisher Scientific, Waltham, MA, U.S.A.). Various concentrations of H_2_O_2_ solution were added to NaAc buffer and continually stirred until the dissolved oxygen value was stable, followed by addition of PDA@Co_3_O_4_ NPs (100 μg/mL). Dissolved oxygen concentrations were recorded over time, and the initial reaction rates were calculated according to the Michaelis–Menten equation, v = *V_max_* × [S]/(*K_m_* + [S]), where v is the initial velocity, *V_max_* is the maximum reaction velocity, [S] is the concentration of substrate H_2_O_2_, and *K_m_* is the Michaelis constant.

Peroxidase (POD)-like activities of PDA@Co_3_O_4_ NPs and bare Co_3_O_4_ NPs were assessed by measuring the oxidation of 3,3’,5,5´-tetramethylbenzidine (TMB) in the presence of H_2_O_2_. Typically, PDA@Co_3_O_4_ NPs or bare Co_3_O_4_ NPs (both at 100 μg/mL) were added to NaAc buffer (0.1 M, pH 4.0) containing TMB (0.5 mM), followed by incubation for 5 min at RT. The blue color intensity was recorded at 652 nm using a microplate reader (Synergy H1). Oxidase (OXD)-like activities were measured following the same procedure as in the POD-assay but in the absence of H_2_O_2_. Superoxide dismutase (SOD)-like activities were evaluated by measuring the reduction in cytochrome C at 550 nm in the presence of superoxide radicals (O_2_^•−^). Xanthine and xanthine oxidase (XO) were used as the source to generate O_2_^•−^. Briefly, PDA@Co_3_O_4_ NPs (100 μg/mL) was added to phosphate buffer (0.05 m, pH 7.4) containing xanthine (0.05 mM), XO (0.15 mU/mL), and cytochrome C (0.01 mM), followed by incubation for 10 min at RT in the dark. The resultant solutions were centrifuged, and used to monitor the absorbance intensities at 550 nm using a microplate reader (Synergy H1).

### 2.4. Detection of S^2−^ Using PDA@Co_3_O_4_ NPs

Detection of S^2−^ using the catalase-like activity of PDA@Co_3_O_4_ NPs was performed as follows. First, aqueous solutions containing various concentrations of sodium sulfide were prepared as the S^2−^ source (0–600 µM). The S^2−^ sample solutions were added to an assay mixture containing PDA@Co_3_O_4_ NPs (100 μg/mL), H_2_O_2_ (20 mM), and TA (0.625 mM) in NaAc buffer solution (0.1 M, pH 6.0). The reaction mixture was incubated for 20 min at RT under UV irradiation (254 nm), and the resulting fluorescent signals were recorded as aforementioned. The limit of detection (LOD) values were calculated as 3 times of standard deviation (SD) of blank value over the slope of the calibration curve. 

To demonstrate the practical utility of the proposed assay, real tap water samples were spiked with S^2−^. For this assay, tap water samples were first collected from the laboratory and filtered through a syringe membrane with a pore size of 0.45 μm to eliminate undesirable molecules. Then, predetermined amounts of S^2−^ were added to the collected tap water samples to prepare spiked samples containing final S^2−^ concentrations of 50, 100, and 200 µM. Finally, the concentrations of S^2−^ in the spiked tap water samples were determined as described above. To measure the accuracy and reproducibility of the assay, we calculated the recovery rate (recovery rate (%) = measured value/actual value × 100) and the coefficient of variation (CV (%) = SD/average × 100), from the six independent assay results.

## 3. Results and Discussion

### 3.1. Synthesis of PDA@Co_3_O_4_ NPs as an Efficient Catalase Mimic to Detect S^2−^

The procedure used for DA-u of DA to form a PDS shell on the Co_3_O_4_ NPs, would have enhanced catalase-like activity with help of additional radical scavenging activity of the PDA shell. Based on this enhanced activity, a highly sensitive system for S^2−^ detection was developed. The high catalase-like activity of PDA@Co_3_O_4_ NPs induced inhibition of the formation of fluorescent 2-hydroxyl TA in the presence of H_2_O_2_, yielding a very low fluorescence background. Importantly, S^2−^ in the sample solution selectively interacted with the surface of PDA@Co_3_O_4_ NPs, resulting in a significant reduction in their catalase-like activity and concomitant increase in fluorescence via the facilitated formation of 2-hydroxyl TA. Specifically, S^2−^ was predicted to be adsorbed on the surface of PDA@Co_3_O_4_ NPs, causing them to aggregate and leading to mass transfer limitations. These factors significantly reduced the catalase-like activity of PDA@Co_3_O_4_ NPs and increased the fluorescence signal proportionally to the amount of target S^2−^ (Figure 1).

### 3.2. Characterization of PDA@Co_3_O_4_ NPs

Structural characteristics of PDA@Co_3_O_4_ NPs were analyzed and compared with those of bare Co_3_O_4_ NPs without a PDA shell by TEM and SEM images. Bare Co_3_O_4_ NPs had spherical shape with 20.03 ± 2.82 nm diameter, calculated from their TEM images, and importantly, relatively thick (~15 nm) shells were clearly observed outside the core NPs from the PDA@Co_3_O_4_ NPs (Figure 2a,b). In basic conditions, DA is known to easily interact with the surface of the NPs by a variety of interactions, including electrostatic interaction, metal coordination, and hydrogen bonding, and induce its polymerization among another DA monomers [25,26,27,28,29,30]. Thus, it was believed that the layer around the Co_3_O_4_ NPs was PDA shell, as also clearly observed in the SEM images (Appendix A in Appendix A). As the concentrations of DA increased, the thickness of polymeric shell concomitantly increased, which was similar to the previous studies (Appendix A) [25,31]. High-resolution TEM (HRTEM) imaging and selected area electron diffraction imaging (SAED) demonstrated the presence of crystalline Co_3_O_4_ in PDA@Co_3_O_4_ NPs, which fit well with the reported data (JCPDS no. 76–1802) (Figure 2c,d). EDX images also proved the presence of Co, N, and O, which were well distributed throughout the material (Figure 2e). The elemental composition ratios within the PDA@Co_3_O_4_ NPs are provided in Appendix A. 

XRD, FT-IR, and XPS analyses were additionally performed to characterize the synthesized PDA@Co_3_O_4_ NPs in detail. The XRD patterns clearly confirmed the presence of crystalline Co_3_O_4_, and the peaks of PDA@Co_3_O_4_ NPs kept nearly the same intensity compared with those of bare Co_3_O_4_ NPs, proving that the PDA layer does not negatively affect the crystalline structure of core Co_3_O_4_ NPs (Figure 3a). The FT-IR spectra confirmed the chemical structure of PDA@Co_3_O_4_ NPs, with the peaks corresponding to C-O stretch (1295 cm^−1^), N-H stretch (1510 cm^−1^), C-H stretch (around 3000 cm^−1^), and aromatic ring (1605 cm^−1^), which demonstrated the presence of the PDA layer on the surface of Co_3_O_4_ NPs (Figure 3b). The PDA peaks around 3400 cm^−1^, which corresponded to the hydrogen bonds of O-H and N-H, were shifted in PDA@Co_3_O_4_ NPs, proving the interaction between Co_3_O_4_ NPs and catechol hydroxyl group of PDA [24]. Moreover, XPS analysis revealed peaks corresponding to C, N, O, and Co at 283.37, 398.23, 530.34, and 779.3 eV, respectively (Appendix A). The appearance of C, N, and O elements was attributed to the presence of PDA on the Co_3_O_4_ NP surface. In addition, the electronic configurations of the O and Co peaks supported the presence of Co_3_O_4_ NPs (Figure 3c,d) [31]. All these characterizations confirm that PDA@Co_3_O_4_ NPs were successfully formed by incorporation of a PDA layer on crystalline Co_3_O_4_ NPs.

### 3.3. Evaluation of the Catalase-like Activity of PDA@Co_3_O_4_ NPs

Catalase-like activities of PDA@Co_3_O_4_ NPs and control Co_3_O_4_ NPs were evaluated via the decomposition of H_2_O_2_ by monitoring the changes in the fluorescent intensities of TA. In the absence of catalase mimics, H_2_O_2_ under UV irradiation produced hydroxyl radicals which further reacted with TA, generating highly fluorescent 2-hydroxy TA. PDA@Co_3_O_4_ NPs or bare Co_3_O_4_ NPs catalyzed the decomposition of H_2_O_2_ to H_2_O and O_2_, resulting in a decrease in the fluorescent signal, and importantly, PDA@Co_3_O_4_ NPs exhibited much higher activity, which was up to approximately fourfold higher than that of bare Co_3_O_4_ NPs (Figure 4a,b). The thickness of the PDA layer significantly affected the catalase-like activity of the materials (Figure 4b). PDA@Co_3_O_4_ NPs with 2 mg/mL DA (2-PDA@Co_3_O_4_ NPs) and PDA@Co_3_O_4_ NPs with 1 mg/mL DA (1-PDA@Co_3_O_4_ NPs) exhibited higher activity than that of PDA@Co_3_O_4_ NPs with 0.5 mg/mL DA (0.5-PDA@Co_3_O_4_ NPs), and the activity difference between 2-PDA@Co_3_O_4_ NPs and 1-PDA@Co_3_O_4_ NPs was not significant. Thus, 1-PDA@Co_3_O_4_ NPs were chosen and used for further studies. We also investigated the other oxidoreductases (POD, OXD, and SOD)-like activities of PDA@Co_3_O_4_ NPs and bare Co_3_O_4_ NPs (Appendix A). Both PDA@Co_3_O_4_ NPs and bare Co_3_O_4_ NPs were unable to remove the O_2_^•−^ produced by the xanthine and XO reaction. In terms of POD- and OXD-like activity, the PDA@Co_3_O_4_ NPs, unlike Co_3_O_4_ NPs, could not oxidize TMB to produce blue-color product (oxidized TMB), which can be measured at 652 mm. It indicated that the developed PDA@Co_3_O_4_ NPs almost solely exhibited high catalase-like activity, which is beneficial for their utilization in catalase-mediated applications.

Several parameters affecting the activity, such as the concentrations of PDA@Co_3_O_4_ NPs, reaction pH, and temperature were examined to obtain the optimal reaction conditions (Appendix A). With increasing concentrations of PDA@Co_3_O_4_ NPs, the fluorescence intensity gradually decreased, and 100 μg/mL of PDA@Co_3_O_4_ NPs was selected for further experiments (Appendix A). Similar to natural catalase, the activity of PDA@Co_3_O_4_ NPs was dependent on the reaction pH and temperature, and pH 6 and RT were found to be the optimal assay conditions (Appendix A). PDA@Co_3_O_4_ NPs showed high activity (over 60%) over broad pH and temperature ranges, whereas natural catalase did not show considerable activity (below 40%) under harsh conditions (acidic or basic pH, and high temperature over 60 °C). This difference may have resulted from the coated PDA layer, which shows additional catalase-like activity even under harsh reaction environments [32]. 

Under the optimized conditions, stabilities of PDA@Co_3_O_4_ NPs, depending on the pH, temperature, and storage time at RT, were assessed and compared with those of natural catalase. As expected, under all conditions, PDA@Co_3_O_4_ NPs clearly showed improved stabilities, maintaining over 90% of their initial activity, while natural catalase lost over half of its activity under harsh conditions (acidic pH below 4, temperature over 50 °C, and storage over 15 days) (Figure 4c–e). The clear improvement in the stability of PDA@Co_3_O_4_ NPs is beneficial for their practical applications. 

Steady-state kinetic assays of PDA@Co_3_O_4_ NPs were performed to determine the Michaelis constant (*K_m_*) and maximal reaction velocity (*V_max_*), which are important to elucidate reaction mechanism [33]. According to the Michaelis–Menten curve obtained using different H_2_O_2_ concentrations and the corresponding Lineweaver–Burk plot, the kinetic parameters were calculated and compared with those of previously reported values from other Co_3_O_4_-based catalase mimics and natural catalase (Appendix A and Appendix A). The *K_m_* value of PDA@Co_3_O_4_ NPs was 22.1 mM, which was over twofold lower than that of natural catalase and among the lowest values reported for Co_3_O_4_-based catalase mimics. These outcomes suggest that PDA@Co_3_O_4_ NPs have higher affinity toward the substrate H_2_O_2_ compared with that of catalase and most Co_3_O_4_-based nanozymes, possibly because of the PDA shell. The *V_max_* of PDA@Co_3_O_4_ NPs was lower than that of natural catalase but higher than those of recently reported Co_3_O_4_ nanozymes. These observations indicate that combining PDA and Co_3_O_4_ NPs enhanced the catalase-mimicking performance of Co_3_O_4_ NPs.

### 3.4. Analytical Capabilities of PDA@Co_3_O_4_ NPs for the Detection of Sulfide Ion

PDA@Co_3_O_4_ NPs with enhanced catalase-like performances were utilized to fluorescently detect environmentally harmful S^2−^. In the absence of S^2−^, TA-mediated fluorescence was significantly decreased because of the high catalase-like activity of PDA@Co_3_O_4_ NPs (Figure 5a). In the presence of S^2−^, the fluorescence was clearly restored due to the significant reduction in the activity of PDA@Co_3_O_4_ NPs. The S^2−^-mediated reduction in activity may have occurred because of the interaction of S^2−^ on the PDA surface and subsequent inhibition of H_2_O_2_ decomposition, resulting in an increased TA-mediated fluorescence signal. The PDA@Co_3_O_4_ NPs-based S^2−^ biosensing system showed high selectivity for S^2−^ (50 μM), while diverse interfering compounds such as small molecules (glucose, urea), biothiols (glutathione, cysteine), and common ions (Mg^2+^, NH_4_^+^, Ca^2+^, Cl^−^) did not have any considerable signal (below the threshold line), even at tenfold higher concentrations (Figure 5b), confirming that the biosensing strategy selectively detected target S^2−^. On increasing the concentrations of S^2−^, fluorescence intensity gradually increased (Appendix A). From the analysis of dose–response curves, the LOD was calculated as low as 4.3 µM with the linear range up to 200 µM (Figure 5c,d), which is sufficient for practical S^2−^ biosensing in the field [34,35].

We speculated that the possible mechanism of the PDA@Co_3_O_4_ NP-mediated S^2−^ biosensing system was that the target S^2−^ aggressively adsorbed on PDA@Co_3_O_4_ NPs and caused substrate transfer limitations via their aggregation, yielding reduced catalase-like activity. To confirm this prediction, the size of the PDA@Co_3_O_4_ NPs was determined in the presence and absence of S^2−^ (Appendix A). The experiments clearly showed that the PDA@Co_3_O_4_ NPs were aggregated in the presence of S^2−^, leading to larger particle sizes (around 800–1000 nm), whereas non-aggregated PDA@Co_3_O_4_ NPs were less than 500 nm in size. This aggregation may reduce the surface area of PDA@Co_3_O_4_ NPs and, thus, reduce the response towards H_2_O_2_ [36].

Finally, to investigate the practical biosensing capability of the developed system, the PDA@Co_3_O_4_ NP-based assay was used to determine S^2−^ in spiked tap water samples, prepared at three concentrations of S^2−^ (50, 100, 200 µM). The biosensor quantified S^2−^ in tap water with good precision and accuracy, with CVs from 3.56 to 6.67% and recovery from 99.75 to 102.43% (Table 1), validating the excellent reproducibility and reliability. These results suggest that the PDA@Co_3_O_4_ NP-based fluorometric biosensor can be used as an analytical system for the determination of S^2−^ in real aqueous environments.

## 4. Conclusions

We demonstrated that PDA@Co_3_O_4_ NPs are efficient catalase-like nanozymes, with competitive catalytic activity and stability compared with natural catalase and recently reported catalase-like nanozymes. We also proved that S^2−^ induced highly selective inhibition of the catalase-like activity of the PDA@Co_3_O_4_ NPs, presumably due to the increased mass transfer limitation through aggregation. Based on the phenomena, S^2−^ was determined with high selectivity and sensitivity, and was quantified in real tap water with sufficient detection precision. This study provides an efficient approach for developing highly efficient nanozymes using simple surface engineering and nanozyme-mediated biosensors. These nanozymes show significant potential for use in diverse biotechnological applications.

## Figures and Tables

**Figure 1 biosensors-12-01047-f001:**
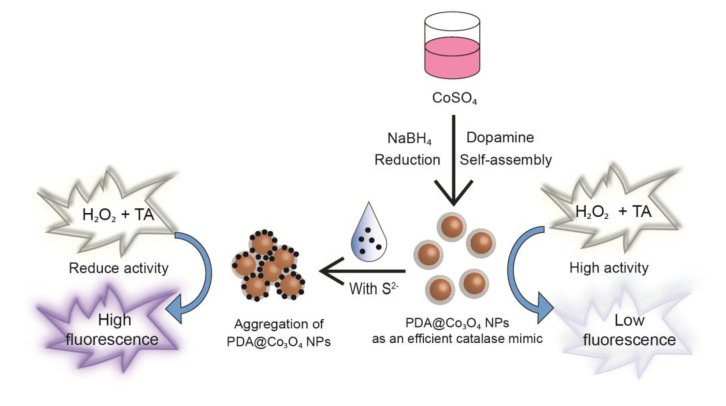
Schematic illustration of the synthesis of catalase-like PDA@Co_3_O_4_ NPs and their application to fluorescently detect sulfide ion (S^2−^).

**Figure 2 biosensors-12-01047-f002:**
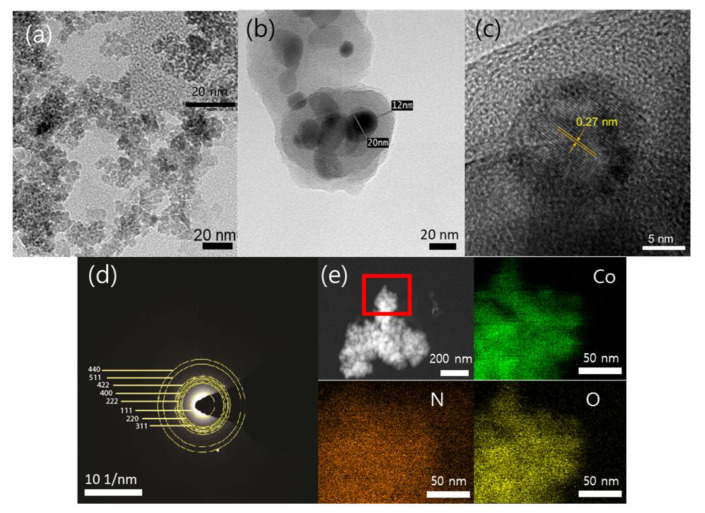
TEM images of (**a**) bare Co_3_O_4_ NPs and (**b**) PDA@Co_3_O_4_ NPs. PDA@Co_3_O_4_ NPs were additionally analyzed by (**c**) HRTEM, (**d**) SAED, and (**e**) EDX mapping images of the selected region (shown as red rectangle).

**Figure 3 biosensors-12-01047-f003:**
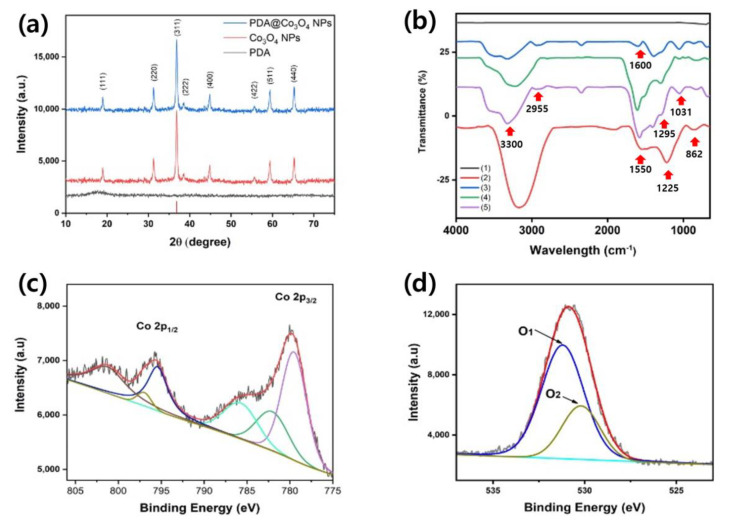
(**a**) XRD spectra, (**b**) FT-IR spectra of bare Co_3_O_4_ NPs, PDA, 0.5−PDA@Co_3_O_4_ NPs, 1−PDA@Co_3_O_4_ NPs, and 2−PDA@Co_3_O_4_ NPs (denoted as 1, 2, 3, 4, and 5, respectively), and high-resolution XPS spectra of PDA@Co_3_O_4_ NPs for (**c**) Co 2p and (**d**) O 1s.

**Figure 4 biosensors-12-01047-f004:**
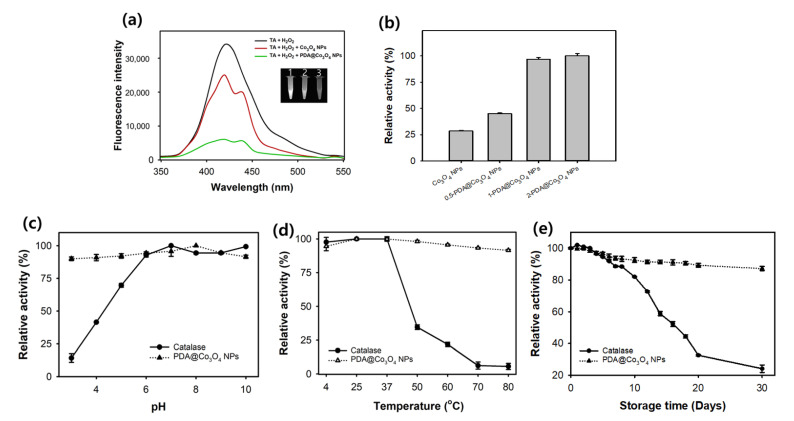
Evaluation of the catalase-like activity of PDA@Co_3_O_4_ NPs. (**a**) Catalase-like activity of bare Co_3_O_4_ NPs (red line) and PDA@Co_3_O_4_ NPs (green line). Insert fluorescent image indicates (1) control, (2) bare Co_3_O_4_ NPs, and (3) PDA@Co_3_O_4_ NPs. (**b**) Comparison of the catalase-like activity among 0.5-PDA@Co_3_O_4_ NPs, 1- PDA@Co_3_O_4_ NPs, and 2-PDA@Co_3_O_4_ NPs. Comparisons of the stability between PDA@Co_3_O_4_ NPs and natural catalase regarding (**c**) pH, (**d**) temperature, and (**e**) storage time at RT.

**Figure 5 biosensors-12-01047-f005:**
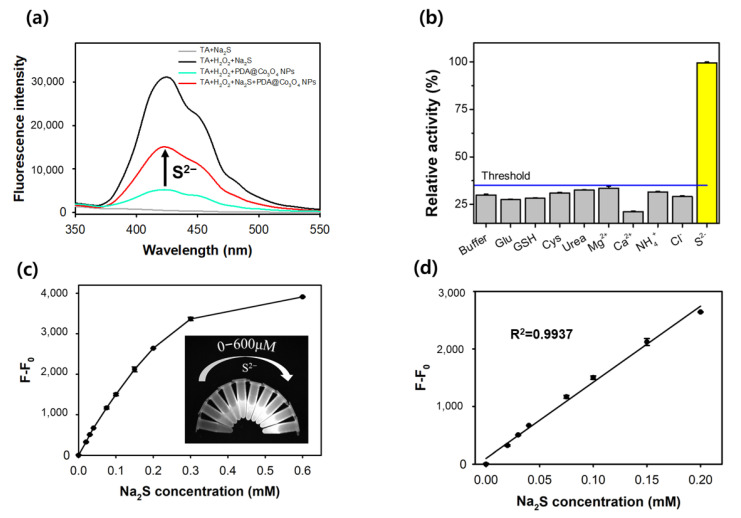
Analytical capabilities of PDA@Co_3_O_4_ NPs for the detection of S^2−^. (**a**) Fluorescence spectra for the inhibition of catalase-like activity of PDA@Co_3_O_4_ NPs by S^2−^. (**b**) Selectivity of PDA@Co_3_O_4_ NPs-based biosensor toward S^2−^. (**c**) Dose–response curve with real florescent images and (**d**) its corresponding linear calibration plot for the determination of diverse concentrations of S^2−^ using the PDA@Co_3_O_4_ NPs–based biosensor.

**Table 1 biosensors-12-01047-t001:** Detection precision of PDA@Co_3_O_4_ NPs-based biosensor for the quantitative determination of S^2−^ spiked in real tap water samples.

Compound	Spiked Level (µM)	Measured ^a^ (µM)	Recovery ^b^ (%) (n = 3)	CV ^c^ (%)
Na_2_S	50	53.16	102.43	3.56
	100	98.23	99.75	3.76
	200	196.24	100.59	6.67

^a^ Mean value of five independent measurements. ^b^ Measured value/expected value × 100. ^c^ Coefficient of variation (CV) = (SD/mean) × 100.

## Data Availability

Not applicable.

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
