# Peer review of "Polydopamine-Coated Co3O4 Nanoparticles as an Efficient Catalase Mimic for Fluorescent Detection of Sulfide Ion"

_biosensors, 2022, doi:10.3390/bios12111047_

Round 1
Reviewer 1 Report
In this research, the authors were displayed the fluorescent sensor by using polydopamine-coated Co3O4 nanoparticles for sulfide ion detection. Adequate physical characterization studies such as SEM, TEM, XRD, FT-IR, and XPS studies were performed. Though some additional improvements are need to meet the journal standard.
Comments:
1. What does PDA bring to this research and what is interaction between PDA and Co3O4?
2. In figure 2B, how the authors estimate the Co3O4 size? Have they performed statistical analysis?
3. Authors need to provide the clear EDX figure with elemental composition percentage.
4. I think that from FT-IR is difficult to highlight that the PDA is strongly bonded on Co3O4, did the authors have observed some shift in the vibration peaks? The presence of the typical bands of PDA and Co3O4 shows that the two components are present in the mixture but it is not a finding of the existence of interactions.
5. Authors need to cites reference from the latest literatures.
6. The English writing needs to be improved.
Reviewer 2 Report
PDA shelled Co3O4 nanoparticles were synthesized to determine the concentration of sulfide iron based on the fluorescence change, whose performance has been compared with that of the pure Co3O4 nanoparticles. The low detection limit was demonstrated as 4.3 μM. Major revision is recommended for its possible publication in Biosensors. Here are my comments:
1. The structures should be re-arranged clearly, for example (in my opinion), to include the characterization images and XRD results in section 2 and analyze the enzyme-like activity performance in Section 3;
2. In figure 3, the numbers should be added on the vertical axis, although the intensity or transmission have been normalized or in the unit of %;
3. In figure 5b, the selectivity of the proposed sensor is not good enough (100% compared to ~25%), are there any suggestions or ideas to improve the selectivity performance. Please discuss this issue;
4. The fluorescence spectra for different concentration of Na2S should be given;
5. How about the response time for the tap water detection?
Reviewer 3 Report
The article title“Polydopamine-coated Co3O4 nanoparticles as an efficient catalase mimic for fluorescent detection of sulfide ion” was reviewed. Both the work and the results were interesting. In this manuscript, PDA@Co3O4 NPs were synthesized by surface engineering and as a highly active catalase-mimicking nanozyme, they were applied for fluorescent detection of S2-. This manuscript can be accepted for publishing in Biosensors “Special Issue: Feature Issue of Biosensor Materials Section” but I have some major remarks before it can be published.
1. Does 1-PDA@Co3O4 NPs in line 228, and 0.5-PDA@Co3O4 NPs line 229 have the same meaning as 2-PDA@Co3O4 NPs (PDA@Co3O4 NPs with 2 mg/mL DA)? Need to add a note.
2. Did you use 1-PDA@Co3O4 NPs when studying” Several parameters affecting the activity, such as concentrations of PDA@Co3O4 NPs, reaction pH, and temperature were examined to obtain the optimal reaction conditions (Figure S5)” and subsequent stability, etc.?
3. Please check the English expression of this article carefully again.
Round 2
Reviewer 1 Report
The authors have performed the revision work in an acceptable way. Now, this revised manuscript can be published in the Biosensor Journal.
Reviewer 2 Report
All comments have been replied.